# The Edge of Unknown: Postoperative Critical Care in Liver Transplantation

**DOI:** 10.3390/jcm11144036

**Published:** 2022-07-12

**Authors:** Fuat H. Saner, Dieter P. Hoyer, Matthias Hartmann, Knut M. Nowak, Dmitri Bezinover

**Affiliations:** 1Department of General-, Visceral- and Transplant Surgery, Medical Center University Duisburg-Essen, 45147 Essen, Germany; dieter.hoyer@uk-essen.de (D.P.H.); knut.nowak@uk-essen.de (K.M.N.); 2Department of Anaesthesia and Critical Care, Medical Center University Duisburg-Essen, 45147 Essen, Germany; matthias.hartmann@uni-due.de; 3Department of Anesthesiology and Perioperative Medicine, Penn State Hershey Medical Center, Penn State College of Medicine, Hershey, PA 17033, USA; dbezinover@pennstate-health.psu.edu

**Keywords:** liver transplantation, critical care management, hemostasis, graft dysfunction

## Abstract

Perioperative care of patients undergoing liver transplantation (LT) is very complex. Metabolic derangements, hypothermia, coagulopathy and thromboses, severe infections, and graft dysfunction can affect outcomes. In this manuscript, we discuss several perioperative problems that can be encountered in LT recipients. The authors present the most up-to-date information regarding predicting and treating hemodynamic instability, coagulation monitoring and management, postoperative ventilation strategies and early extubation, management of infections, and ESLD-related pulmonary complications. In addition, early post-transplant allograft dysfunction will be discussed.

## 1. Introduction

Liver transplantation (LT) as a treatment for end stage liver disease (ESLD) became a routine procedure in many centers after the first success in 1967. Initially, a relatively low postoperative survival was primarily related to the lack of effective immunosuppression, surgical technique, and a limited understanding of ESLD-related coagulation pathophysiology. This often resulted in acute rejection, primary graft non-function (PNF), uncontrolled bleeding with massive transfusion, severe thromboses, and sepsis.

Advances in pre-transplant evaluation, surgical technique, and an increased understanding of the pathophysiology of cirrhosis significantly improved patient outcome. Due to the increasing demand for organs, however, a higher number of extended criteria grafts (ECD) are currently being used for transplantation. The use of ECD grafts has been shown to be associated with a higher rate of early allograft dysfunction (EAD), which results in dysfunction of other organ systems.

In the last twenty years, the demographics of patients needing LT has changed. Candidates are now often older, more deconditioned, and frailer. A significant number of patients require care in an intensive care unit (ICU) before LT for preexisting conditions such as infections and sepsis, dialysis-dependent renal failure, and vasopressor support that can subsequently complicate intra- and postoperative care, even if a high-quality graft is used. Even when patients have been discharged from the ICU postoperatively, around 20% of patients require readmission to the ICU, primarily for cardio-pulmonary complications [1].

This review will principally discuss postoperative care in the ICU with a focus on cardio-pulmonary function, coagulation, and early allograft dysfunction. 

## 2. Hemodynamics: Perioperative Challenges

### 2.1. Restrictive Fluid Management

ESLD is associated with significant changes in hemodynamics in both the systemic and portal circulation [2,3]. It has been demonstrated that depending on the model of end-stage liver disease (MELD) score, up to 60% of LT candidates have a significant drop in mean arterial pressure (MAP), intraoperatively requiring treatment with vasopressors [4]. The reason for this significant hypotension is increased nitric oxide production with subsequent activation of cGMP resulting in profound vasodilation [5]. Fluid administration results in increased portal pressure and liver congestion [2]. Most recent published papers favor a restrictive fluid management strategy, which is associated with a reduction in the amount of blood transfused [6].

Massive transfusion leads to increased hydrostatic pressure, which can result in liver congestion and pulmonary edema [7]. The prevalence of pulmonary edema after LT can be as high as 50% [8]. When this occurs, it is associated with prolonged postoperative ventilation and increased length of stay in the ICU [9]. In a prospective randomized double-blind, placebo-controlled study, Ponnudurai at et al. demonstrated that restrictive fluid management in combination with vasopressor support was associated with a reduction in re-intubation and ventilation-associated morbidity [10].

### 2.2. Central Venous Pressure

Central venous pressure (CVP) is frequently used to assess fluid status and is almost always elevated in cirrhotic patients. CVP is directly related to hepatic venous pressure and almost linearly correlates with the amount of blood loss [11]. Maintenance of a low CVP (≤5 mbar) and preoperative CVP reduction by phlebotomy have been reported to be beneficial in reducing blood loss during hepatectomy or LT [12,13]. These results may be questionable. Most of these studies were performed years ago. Since then, both surgical and anesthetic management have changed significantly [14]. In addition, CVP values during major abdominal surgery are not always accurate, and can be artificially elevated due to pressure from surgical retractors [15].

### 2.3. Myocardial Injury

Myocardial injury following non-cardiac surgery (MINS) is caused by an imbalance in myocardial oxygen supply and demand. The diagnosis is based on elevated troponin T(Tni) levels in the absence of electrocardiogram (ECG) changes and myocardial symptoms [16].

In a single-center study with 1386 LTs, 502 patients had an increased TnI within 30 days following LT. The prevalence of MINS in this group was 40%. The 30-day mortality rate was higher in the MINS group (11.8%) compared to the non-MINS group (3.3%).

In this study, several preoperative factors (higher MELD score, need for dialysis, preoperative intubation, and variceal bleeding) and several intraoperative factors (hemodynamic instability, reperfusion syndrome, requirement of vasopressors, and rate of transfusion) were associated with MINS. Interestingly, a history of coronary artery disease (CAD) was not associated with MINS in this study [17]. 

### 2.4. Cardiomyopathy

The prevalence of cardiomyopathy and heart failure (defined as a decrease in left ventricular ejection fraction) has been reported to be between 3 and 7% [18,19] in LT recipients. The management of heart failure in LT patients does not differ from typical management. Although specific diagnostic criteria for cirrhotic cardiomyopathy have been published [20], specific recommendations for management of this condition are not yet available [21]. Takotsubo syndrome (TTS), atypical myocardial ballooning, is a cardiomyopathy characterized by the development of acute, severe left ventricular dysfunction triggered by catecholamine excess and surgical stress (such as that seen with LT). In a large cohort, TTS occurred in 1.7% of patients after LT [22]. During the acute phase, TTS presents with hemodynamic and/or conduction abnormalities, which can result in cardiogenic shock. TTS can mimic a myocardial infarction in its initial presentation without angiographic evidence of coronary artery disease [23].

### 2.5. Atrial Fibrillation

Atrial Fibrillation (AF) is associated with significant morbidity and mortality, including stroke and heart failure. The prevalence and impact of AF on patient outcome after non-cardiac surgery has been well documented and is about 3% [24]. The prevalence of perioperative AF in patients undergoing LT is higher than in the general population. In a meta-analysis, Chokesuwattanaskul et al. demonstrated that the prevalence of pre-existing AF in LT candidates is over 5%, and in transplanted patients, exceeded 8% [25]. AF is likely associated with severity of ESLD. In their study, nearly a third of patients with MELD scores of 32 or higher developed AF [26]. In a single-center series of 717 consecutive LTs, 32 patients (4.5%) had documented AF prior to surgery. Compared to an age-matched control group, patients with AF had a higher prevalence of adverse cardiovascular events during both intra- and postoperative periods. Overall graft and patient survival, however, were similar between groups [27]. In another single-center evaluation of 757 LT patients, 19 (2.5%) had documented preoperative AF. Compared to non-AF patients, patients with AF had lower 30-day (84% vs. 97%) and one-year (68% vs. 90%) survival [28]. In 1387 consecutive LT patients, the prevalence of postoperative AF within 30 days of surgery was 7.4%. Patients with postoperative AF were older, had a higher MELD score, and were more likely to require preoperative intubation, dialysis, and vasopressor support. Overall mortality and graft failure were significantly higher in patients with postoperative AF, as well as the incidence of postoperative renal failure and duration of hospitalization. Management is aimed at treating the underlying cause.

AF is not a contraindication for LT; however, the higher incidence of associated perioperative complications must be taken into consideration.

Common cardiovascular disorders in patients with ESLD are presented in Figure 1.

## 3. Mechanical Ventilation after Transplantation

### 3.1. Early Postoperative Extubation

Prolonged mechanical ventilation after LT is associated with worse outcomes [29,30,31]. The first study describing the early benefits of extubation after LT in combination with restrictive fluid management was published in 1990 by Rossaint et al. [32]. In 1997, Mandell et al. evaluated the results of an early extubation trial performed in two large US LT centers. The authors found that extubation after LT is safe and cost-effective in selected patients. Biancofiore et al. demonstrated that early extubation (within 3 h after surgery) was feasible and safe in a large cohort of transplant recipients (*n* = 181 cases) without specific patient pre-selection [33]. Among these patients, however, 13.2% required re-intubation. Another multicenter study evaluated the feasibility of extubation within 1 h after surgery [34]. Only 7 of 391 patients (1.8%) required re-intubation. A recent systematic review investigated the quality of evidence for early extubation after LT as a part of an enhanced recovery after surgery (ERAS) project performed by the International Liver Transplantation Society (ILTS) [35]. They found that, overall, complications were reduced in the ERAS cohort versus controls (OR = 0.4 (CI 0.2, 0.7)), with no significant differences in mortality or hospital readmission rates. ICU unit and hospital length of stay were shorter in the ERAS group.

Although these studies demonstrate that immediate postoperative extubation is safe in LT recipients, the patients in these studies had relatively low MELD scores (below 18). This indicates that the extubation protocols used for these patients are likely not directly applicable for patients with more advanced ESLD.

Once the decision to extubate is made, strategies to avoid reintubation should be prioritized. The use of non-invasive positive pressure ventilation (NIPPV) should be considered. It has been demonstrated that use of NIPPV was able to reduce the rate of reintubation when respiratory failure occurred immediately after extubation [36]. Factors that limit the use of NIPPV include altered mental status, shock, multi-organ failure, and extreme frailty. For these patients, the use of high-flow nasal cannula (HFNC) can be considered. Equivalent efficacy for HFNC and NIPPV in patients at high risk for reintubation has recently been demonstrated [37]. 

### 3.2. Pulmonary Complications 

The incidence of pulmonary complications in the early postoperative period after LT has been reported to be up to 50% [38,39].

Acute respiratory distress syndrome (ARDS) is characterized by respiratory failure not caused by cardiac failure, bilateral opacities on chest imaging, and oxygenation impairment with a Horovitz (oxygenation) index below 200 mmHg [40]. Overall mortality depends on the severity of ARDS and ranges from 20–50% [40]. Risk factors for ARDS in LT patients include massive transfusion, fluid overload, sepsis, and aspiration [41]. The incidence of ARDS after LT varies between centers. In one study, the rate of ARDS in LT patients was 4.1% (71/1726 patients) [42]. Most cases were characterized as mild to moderate, and occurred on the first postoperative day. Patients with ARDS, however, had a 2-fold increase in 1-year mortality.

The treatment of ARDS in patients having LT is no different from treating ARDS in other groups of patients. There is strong evidence that lung protective ventilation with 6 mL/kg improves outcome [43]. Increased intrathoracic pressure due to high PEEP may impede venous outflow [44]. However, in 2006, a group from Essen published a cohort study of 65 LT patients assigned to three different PEEP levels (0, 5, and 10 mbar) [45]. At all three PEEP levels, there was no change in flow velocities in the hepatic artery, portal vein, or hepatic veins. This demonstrated that, at least at these PEEP levels, hepatic perfusion was not impaired. In a later study performed by the same group, it was demonstrated that even long-term ventilation using higher PEEP levels (at least 4 days with PEEP up to 15 mbar) did not impair liver function [46]. These studies demonstrate that PEEP can be used in the postoperative period as needed.

Pleural effusions (mainly right sided) are common after LT, and can be new or pre-existing. Placement of a chest tube and pleural drainage may help weaning from the ventilator, but supporting data are lacking. Most effusions are minimally symptomatic and self-limited, and do not require any interventions.

## 4. Infection Prophylaxis

Infection prevention is an important problem in the care of LT recipients. The incidence of infection after LT varies from 53% to 79%, with most infections occurring in the first month after transplantation [47].

In 2009, a survey was sent out to all LT centers who are members of the European Liver and Intestine Transplant Association [48]. This survey analyzed the differences in prophylactic antimicrobial regimens used in LT recipients. For elective LT, beta-lactam antibiotics or co-trimoxazole were used as first-line antibiotic prophylaxis in 25% of all centers; third or fourth generation cephalosporins, as well as glycopeptide, carbapenem, or antipseudomonas were used in 73% of centers; and 2% of centers used a 6-month rotation strategy using two different types of broad-spectrum antibiotics. Antifungal prophylaxis was administered in 35% of centers for all LT recipients; only in patients at risk in 53% of centers; and in 12% of centers, antifungal prophylaxis was not used. The duration of antibiotic prophylaxis was also different between centers, from 24 h to 1 week.

In 2019, Berry et al. published a randomized controlled trial (RCT) comparing 72 h of perioperative antibiotic prophylaxis protocols [49]. Their initial hypothesis was that 72 h preoperative prophylaxis would decrease rates of surgical site infection (SSI) in LT patients when compared with intraoperative antibiotic prophylaxis alone. A total of 102 patients were randomized as follows: 51 patients to the extended antibiotic group, and 51 to the intraoperative antibiotic group. Rates of SSI and nosocomial infection were not different between groups. Moreover, ICU and hospital length of stay (LOS), 30-day mortality, and time to infection were also similar in both groups. Patients developing infections had longer ICU and hospital LOS and a higher prevalence of reoperation. These results suggest that intraoperative antibiotic prophylaxis alone is acceptable for LT without increased risk of infection.

There is a general recommendation to use antifungal prophylaxis in high-risk patients with a MELD score above 20 [50,51]. Antifungal treatment is also recommended for patients with MELD scores above 30, patients needing reoperation (for bleeding or bile leak), on renal replacement therapy, receiving pulsed dose cortisone for rejection, or categorized as at high-risk for fungal infection.

In 2006, Cruciani et al. published a meta-analysis that included six RCT evaluating antifungal prophylaxis in LT recipients [52]. They found a reduced rate of colonization, fungal infection, and fungal-related deaths in the groups where antifungal prophylaxis was performed. Compared to controls, however, the rate of resistant *Candida* spp. was higher in the prophylaxis group; although, the overall mortality was not different. In 2014, Evans at al. published a similar meta-analysis encompassing 14 RCT that included echinocandins (a new drug for prophylaxis) [53]. The results were similar: the use of antifungal drugs as prophylaxis was protective against colonization, invasive fungal infection (IFI), and IFI-related deaths, but overall mortality was not affected.

In a multicenter, retrospective study, Raghuram et al. evaluated the rate of fungal infections in high-volume US LT centers over a period of 5 years [54]. They found that the rate of IFI was 11.5%. The main fungus isolated was *non-albicans* (58%), and only 28% were isolated as *C. albicans*, 15% *aspergillosis,* and 3% *Cryptococcus*. Among the *C. albicans,* only 44% were susceptible to fluconazole. One hundred percent of *C. parapsilosis* were resistant to fluconazole. The authors concluded that the use of antifungal prophylaxis did not reduce the rate of IFI. Moreover, infections with fluconazole-resistant *Candida* spp. were associated with a higher mortality. In another cohort study published in 2008, the incidence of IFI was 6.1% [55]. At that time, all patients received antifungal prophylaxis with fluconazole. Thirteen years later, the same group reported an IFI rate of 5.6%, but without antifungal prophylaxis [56]. It is interesting to note that in the second period study, the patients’ MELD scores were higher (14 vs. 20).

In conclusion, recent studies have demonstrated that prolonged (more than 24 h) antibacterial prophylaxis is not required. Antifungal prophylaxis should be considered in high-risk patients; however, a clear survival benefit has not been demonstrated.

Viral infections are also a significant problem in the postoperative period, with human cytomegalovirus (CMV) being most common in LT recipients. The main risk factor for developing CMV is a recipient’s CMV-seronegative status. Without prophylaxis, the prevalence of CMV has been reported to be 78–88% in seronegative recipients (R-) obtaining a seropositive organ (D+). This incidence decreases to 13% if donor and recipient are CMV-seronegative [57]. Oral valganciclovir and intravenous gancicilor are used for both prophylaxis and treatment [58]. The recommended dose of valganciclovir is 900 mg/day, which should be adjusted when kidney function is impaired. The duration of prophylaxis in high-risk patients (D+/R−) has not been specified, but is generally recommended for 6 months [59]. For R+ recipients, the recommend duration of therapy is 3 months. 

The management of chronic hepatitis B infections (HBV) is very complex and beyond the scope of this review. In the absence of prophylaxis, recurrence of HBV cirrhosis after LT is very high [60]. The use of hepatitis B immunglobulin (HBIG), becoming available only within the last 20 years [61], together with antiviral medications such as lamivudine, entecavir, and tenofovir, improved 5-year survival from 45% to 85% after LT [62]. Based on this success, HBs-Ag-positive or Anti-HBc-positive donor organs are recognized as extended criteria organs and can be used for LT. Prophylaxis with antiviral medication, with or without HBIG, is recommended to prevent transmission of HBV, if these grafts are used for LT [63].

In conclusion, the recommended CMV prophylaxis includes valganciclor or ganciclovir for D(+) and R(−) organs. In patients transplanted due to hepatitis B, or if a HBc-positive donor organ is used, prophylaxis should be performed using HBIG and the antiviral medications, entecavir or tenofovir.

## 5. Management of Coagulopathy

### 5.1. Coagulopathy Assessment

The first attempts at human LT were associated with very high mortality. Uncontrolled bleeding (and in few cases, thromboembolism) were the major causes of death in these patients. This was only partially related to surgical expertise. An even greater problem was a lack of experience in the management of coagulopathy in patients with ESLD. It was quickly understood that standard laboratory tests (SLTs) do not accurately reflect the coagulation profile in patients with ESLD. In 1981, Ewe et al. published a paper evaluating bleeding after liver biopsy [64]. The authors raised a concern that SLTs did not correlate with the bleeding time after procedure. The number of patients in this study with a completely normal coagulation profile had prolonged bleeding, whereas other patients with an international normalized ratio (INR) above 3 did not have significant bleeding. Several patients with a platelet count above >100/nL had prolonged bleeding, and several patients with a platelet count below 20/nL did not bleed. These observations were confirmed in a meta-analysis performed by Haas et al. [65] They evaluated 53 studies related to the overall management of the bleeding (not just in patients with ESLD), and found that SLTs are not useful in guiding bleeding management. SLTs frequently do not correctly reflect the overall coagulation picture in patients with ESLD because they are designed to assess only procoagulants (in patients with ESLD, levels of both pro-and anti-coagulants are decreased). SLTs are measured in plasma (with the addition of thrombin and calcium) resulting from centrifuged citrated blood. This approach does not reflect the interaction between the different branches of the coagulation cascade. Viscoelastic tests (VET) can be used as an alternative to SLTs. As opposed to SLTs, VETs are performed on whole blood and reflect the interaction between pro- and anti-coagulants, and platelets [66].

The use of VET for managing hemorrhagic shock and other bleeding disorders is recommended by the European Society of Anesthesiology [67]. In two RCTs, the use of VET to guide transfusion in patients with ESLD was significantly associated with decreased blood product use without increasing spontaneous or procedure-related bleeding [68,69]. The use of VET for managing coagulopathy during LT was recently recommended in the ERAS project performed by the ILTS [70].

### 5.2. Thromboembolism in ESLD

The coagulation system in ESLD patients is in a delicate balance between bleeding and clotting. It can be easily tilted in either direction [71]. One of the first papers confirming that patients with ESLD are prone to thromboses was based on the Danish National Registry and published in 2009 [72]. The authors evaluated over 99,000 patients with venous thromboembolism (VTE) compared to over 400,000 controls. Patients with both cirrhotic and non-cirrhotic liver disease had an elevated relative risk for thromboses (1.74 and 1.87, respectively). Similar results were demonstrated later in the US [73]. The causes of hypercoagulability in ESLD are mostly related to endothelial dysfunction [74,75], with release of von Willebrand factor (vWF), factor VIII, and plasminogen activator inhibitor-1 [66,76,77,78] from the endothelium combined with a simultaneous decrease in hepatic production of ADAMRS13 (a cleaving protease regulating vWF) [79]. Other causes of hypercoagulability include overproduction of thrombin due to thrombomodulin resistance [80] and increased clot stability [81]. There is also number of genetic mutations predisposing patients with ESLD to thromboses [82,83].

### 5.3. Use of Coagulation Factor Concentrates

As an alternative to fresh-frozen plasma (FFP), coagulation factor concentrates are available. The most commonly used products in hepatic surgery or in cirrhotic patients are prothrombin complex concentrates (PCC), which include factor II, VII, IX, and X. Additionally, four-factor PCC contains heparin, and proteins C and S, which makes this a well-balanced compound. In vitro, PCC has been demonstrated to improve thrombin generation in patients with ESLD significantly better than FFP [84]. The use of factor concentrates appears safe when used in bleeding patients if it is monitored and guided by VET [85,86]. 

### 5.4. Preemptive Management of Coagulation

Hemostasis treatment should only be performed in case of bleeding [87]. In patients with ESLD, bleeding rarely occurs solely due to factor deficiency, and is usually associated with portal hypertension [87]. Volume expansion prompts an increasing portal venous pressure and related risk of bleeding. This is the reason why prophylactic fresh-frozen plasma (FFP) transfusion should be avoided. It has been demonstrated that transfusion of six units of FFPs will increase the portal pressure by 15 mmHg, which correlates well with increased bleeding risk [88]. FFP transfusion has only a limited effect on either correcting factor deficiencies or improving thrombin generation [89,90].

It has also been demonstrated that prophylactic fibrinogen administration in LT recipients did not affect transfusion requirements [91].

A recent paper prepared as a part of an ILTS ERAS project did not recommend the use of prophylactic antifibrinolytics in LT recipients [70]. Antifibrinolytics should be used only in the case of fibrinolysis-related clinical bleeding as diagnosed with VET.

## 6. Thrombotic Microangiopathy

Thrombotic microangiopathy (TMA) is a very complex condition. Clinical signs of TMA include microthrombotic hemolysis, thrombocytopenia, and organ injury [92]. The most frequent manifestation of TMA is thrombotic thrombocytopenic purpura (TTP), first reported in 1924 [93]. Moschcowitz described a case of a 16-year-old girl initially presenting with weakness, fever, and hemiparesis. Her condition deteriorated and she died 14 days after admission. Autopsy demonstrated massive thrombi in arterioles and capillaries in the kidneys. TTP can be hereditary or acquired. Hereditary TTP is due to a mutation in ADAMTS 13, and acquired TTP is associated with autoantibody production against ADAMTS 13. Both lead to an increased concentration of von-Willebrand factor (vWF) [94]. TMA presents as microthrombotic hemolytic anemia with thrombocytopenia. Patients develop kidney failure and neurologic deficits. TMA can also present as an atypical hemolytic-uremic syndrome (HUS). HUS is principally caused by Shiga-toxin producing *E. coli* [95]. Cell damage occurs when Shiga binds to globotriaosylceramide (GB3) on endothelial, mesangial, and tubular cells. This results in an inflammatory reaction and cell apoptosis with subsequent release of vWF from endothelial cells [94]. Atypical HUS is complement-mediated. 

Cyclosporine and tacrolimus may also induce TMA by inhibiting prostacyclin and vascular-endothelia growth factor (VEGF). This results in damage to endothelial cells and TMA in glomeruli. Treatment includes discontinuing cyclosporine or tacrolimus.

Outcomes with TMA after LT have been described by Takatsuki et al. [96]. The authors reported on 98 LDLT patients who developed TMA soon after transplantation. The 1-, 3-, and 5-year survival were 66.9%, 64.6%, and 62.2%, respectively. The only independent risk factor for mortality was dialysis-dependent kidney failure.

## 7. Hepatopulmonary Syndrome

Hepatopulmonary syndrome (HPS) is a vascular disorder characterized by impaired pulmonary gas exchange. It is caused by pulmonary vasodilatation and/or additional anatomic shunts which occur in patients with ESLD or portal hypertension [97]. The prevalence of HPS in cirrhotic patients is up to 32% [98]. The presence of HPS doubles the risk for death of patients on the waiting list [99]. 

The clinical diagnosis of HPS includes hypoxemia with cyanosis, clubbing of the fingers, P_a_O_2_ < 80 mmHg, alveolo-arterial partial oxygen pressure gradient (AaDO_2_) ≥ 20 mmHg, and orthodeoxia (see Figure 2). The diagnosis should be confirmed by contrast-enhanced echocardiography demonstrating macroaggregates >20 µm appearing in the left ventricle after three or more cardiac cycles, or for patients with underlying pulmonary disease with a ^99m^TC macroaggregated albumin scan demonstrating ^99m^TC macroaggregated albumin brain activity exceeding 6%.

There have been conflicting reports on the association between HPS and postoperative mortality [98,100,101,102,103,104]. These evaluations included small numbers of patients. In 2014, Goldberg et al. published the results of a UNOS database evaluation [105]. They included data from 973 patients receiving HPS-related exception points transplanted between 2002–2012. In this study, LT recipients with more severe preoperative hypoxemia had an increased risk for mortality. Unadjusted survival rates post-transplant were 84% for patients with a preoperative p_a_O_2_ between 44.1–54 mmHg, and 68% for those with p_a_O_2_ below 44 mmHg on room air.

Although the syndrome was first described in 1884, besides LT, there is still not an effective treatment [106]. Intraoperative strategies to improve oxygenation include the use of methylene blue and extracorporeal membrane oxygenation (ECMO) [107,108]. 

Improved oxygenation usually does not occur immediately after transplantation. HPS can be associated with severe posttransplant hypoxia, defined as the inability to maintain SaO_2_ > 85% despite FiO_2_ = 100%, and carries a 45% risk for mortality [109]. The time to improved oxygenation cannot be forecast, and may last from several weeks to 1 year. In some cases, oxygen dependence is persistent despite excellent graft function [109]. 

### Porto-Pulmonary Hypertension

Porto-pulmonary hypertension (POPH) in patients with ESLD is caused by pulmonary vasoconstriction, proliferation of endothelium or smooth muscle, and platelet aggregation with release of thromboxane A2 [110]. 

The diagnosis of POPH is based on the presence of portal hypertension in the setting of chronic liver disease with a mPAP ≥ 25 mmHg, pulmonary vascular resistance (PVR) ≥ 240 dyn × s × cm^−5^, and with the exclusion of left ventricular heart failure (pulmonary capillary wedge pressure (PCWP) ≤ 15 mmHg) [110]. The severity of POPH depends on the mean arterial pulmonary pressure (mPAP) assessed by right heart catheter. Mild POPH is defined as 25 mmHg ≤ mPAP < 35 mmHg, moderate as 35 ≤ mPAP ≤ 45 mmHg, and severe when the mPAP > 45 mmHg [111]. Severe POPH is usually associated with right ventricular dysfunction and decreased cardiac output (less than 2 L/min^−1^ m^−2^).

The prevalence of POPH in patients with ESLD is between 5–6% POPH [112]. In a case-control study, female sex and autoimmune hepatitis as the cause for cirrhosis were identified as risk factors for POPH [99]. The authors found that estrogen-signaling could be involved in the development of POPH.

Untreated, POPH is associated with a 1-year survival between 35% and 46% [113,114]. A lower cardiac index or increased right atrial pressure are associated with increased mortality [115,116]. A normal cardiac index (CI) in patients with POPH is a recognized manifestation of significant right heart dysfunction [116].

POPH has more therapeutic options than HPS. Prostacyclin analogues possess vasodilator and antithrombotic effects. A few case reports and case control studies have reported improvement in hemodynamics when epoprostenol was used intravenously [117,118,119]. Favorable short-term effects have been demonstrated when inhaled iloprost was used in severe cases [120].

Phosphodiesterase inhibitors modulate the effect of NO. Sildenafil, the principal medication in this group, has been reported to improve hemodynamics, increase cardiac output, and decrease PVR [121,122,123].

Endothelin receptor antagonists (Bosentan), which are used to treat pulmonary arterial hypertension, are also associated with an improvement of hemodynamics in patients with cirrhosis and POPH [124,125,126]. One study evaluated the long-term effect of Bosentan administration, and found that after 5 months of treatment, PVR was reduced by 31%, whereas CI increased by 31%. [127]. However, in seven cases, there was a 3-fold elevation of transaminases from baseline which could be controlled by dose-reduction. Another medication in this group (Ambrisentan) has been found to be effective in improving hemodynamics and functional status in patients with POPH [128]. The use of Macitentan was associated with a significant decrease in pulmonary vascular resistance without hepatic-related adverse effects [129].

## 8. Allograft Dysfunction

EAD and primary PNF are some of the most difficult complications to manage in a postoperative setting. Both EAD and PNF are associated with hemodynamic instability, AKI, coagulopathy, and cardiac complications.

### 8.1. Early Allograft Dysfunction

The most accepted definition of EAD was recently published by Olthoff in 2010 [130]. This definition minimizes the contribution of an elevated preoperative bilirubin level and perioperative coagulopathy by assessing the parameters on postoperative day (POD) 7. The downside of this definition is its applicability relatively late after LT.

There are several other models to assess graft (dys)function after transplantation that not only include postoperative parameters, but can also assess the severity of EAD. Examples for these assessment tools are the Model for Early Allograft Function Scoring (MEAF) or Liver Graft Assessment Following Transplantation (L-GrAFT) [131,132]. Historical scores to grade early allograft dysfunction (EAD) are presented in Table 1 [130,133,134,135]. Unfortunately, all these definitions are static and do not include the assessment of the entire clinical picture. For example, a patient after LT with an elevated aspartate aminotransferase (AST) and alanine aminotransferase (ALT) above 5000 U/L, but with low or no vasopressor requirement, continuously decreasing lactate level, and preserved renal function, is far less concerning in comparison to a patient with a high vasopressor requirement, impaired lactate clearance, and who is developing renal failure.

The prevalence of EAD after LT is between 6–35%. The association between decreased graft function and patient survival has been demonstrated in all clinical studies. Hoyer et al. found that 30-day and 1-year mortality in patients with EAD was significantly higher in comparison to patients without EAD (31.5% vs. 6.9% and 50.9% vs. 20.6%, *p* < 0.0001, respectively) [136].

In addition to hemodynamic management and treating kidney failure and coagulopathy, the role of liver-supporting systems, particularly, non-biologic liver support systems, are important in managing EAD. Therapeutic plasma exchange can be used in the setting of primary graft non-function, and has resulted in an improvement in EAD [137,138]. Some reports, however, have demonstrated increased mortality and graft loss after the use of plasma exchange [139,140].

A recent study in the living donor LT setting reported higher rates of septic complications, renal replacement therapy, and deaths after plasma exchange in patients with EAD [141]. Therefore, a general recommendation for plasma exchange in the setting of EAD cannot be given and should be decided on a case-by-case basis.

### 8.2. Primary Graft Non-Function

In 1–7% of all LTs, the allografts never gain sufficient function, despite sufficient graft perfusion and without any technical problem. This condition is described as graft PNF and is clinically apparent within the first 7–14 PODs.

An internationally accepted definition of PNF is lacking. Several studies pointed out that elevation of aminotransferases, INR, and bilirubin, as well as lactate levels and acidosis, are relevant in making the diagnosis. It has been suggested to evaluate these blood parameters no earlier than POD 3 to avoid a diagnosis of PNF in error [130,142,143]. Though the threshold levels of aminotransferases used in making the decision to relist range from 1000–10,000 U/L [142], data exists for the non-relevance of aminotransferase levels as a single parameter to base a decision to relist due to PNF [144]. US allocation policies suggests a maximum level of AST > 3000 IU/L for relisting. Data from Eurotransplant from 2010–2016 demonstrated a median AST value of 4655 IU/L and ALT of 2351IU/L in cases of PNF [145]. All other parameters are utilized heterogeneously in different parts of the world [142].

Most of these cases must be re-listed with “high-urgency” status and need early re-transplantation. In the US and UK, a high-urgency status is only possible until POD 7. The disadvantage of this restriction is that re-transplantation might be indicated too early. In the Eurotransplant area, the diagnosis of PNF can be made until POD 14, which helps avoid unnecessary early re-transplantations.

## 9. Conclusions

Perioperative critical care of LT recipients presents a major challenge for anesthesiologists and ICU physicians. Managing these patients requires an in-depth understanding of the pathophysiology of ESLD.

The improvement in survival of patients after LT in the past 60 years is not only linked to better surgical expertise and improvements in immunosuppression, but also a better understanding and management of infections, hemodynamics, renal dysfunction, fluid administration, and coagulation.

In the early postoperative period, the critical-care physician coordinates the care provided by physicians from many other specialties primarily involved in transplantation. Patient outcome depends on a well-functioning multidisciplinary team.

## Figures and Tables

**Figure 1 jcm-11-04036-f001:**
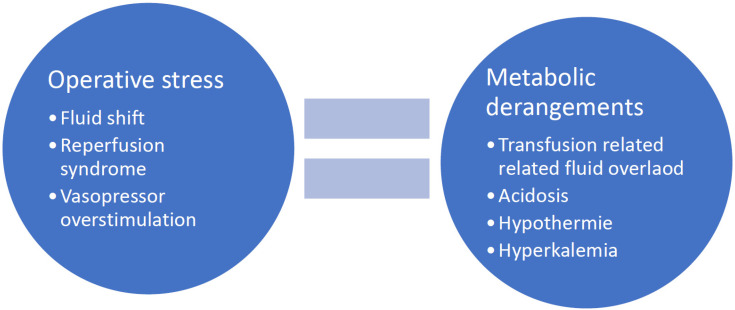
Common causes of postoperative cardiovascular disorder.

**Figure 2 jcm-11-04036-f002:**
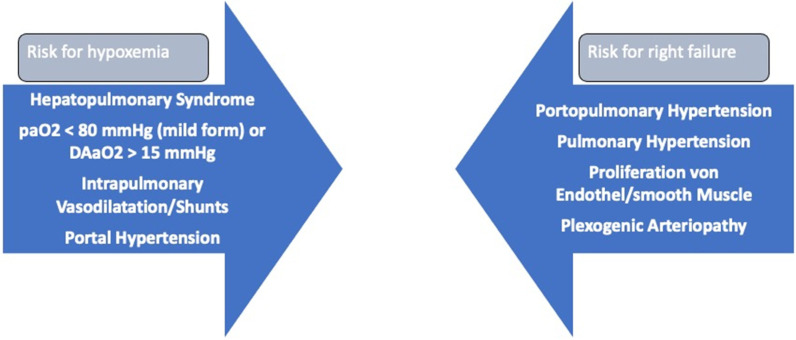
Comparison Hepatopulmonary syndrome and Portopulmonary hypertension.

**Table 1 jcm-11-04036-t001:** Historical definitions of early allograft dysfunction (EAD).

Year	Author	Journal	Term	Lab Values	Others
1993	Ploeg	*Transplantation* [135]	Primary Dysfunction	AST > 2000 IU/L PT > 16 s NH3 < 50 µmol/L from POD 2–7	-
1998	Deschenes	*Transplantation* [134]	Early Allograft Dysfunction	Bilirubin > 10 mg/dL Prothrombin time (PT) ≥ 17 s	Hepatic Encephalopathy
2002	Nanashima	*Transpl. Proc.* [133]		AST or ALT > 1500 IU/L in two consecutive tests within first 72 h	-
2010	Olthoff	*Am. J. Transpl.* [130]	Early Allograft Dysfunction	AST/ALT > 2000 IU/L within 7 PODs Bilirubin ≥ 10 mg/dL INR ≥ 1.6 ON POD 7	-

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
