# Peer review of "The Edge of Unknown: Postoperative Critical Care in Liver Transplantation"

_jcm, 2022, doi:10.3390/jcm11144036_

Round 1
Reviewer 1 Report
Liver transplantation is a life-saving procedure for patients with end-stage liver disease (ESLD), however due to its complexity as an operation as well as due to the major fluid shifts and alteration of the overall physiology of the patient, management of the patient in the postoperative period remains a significant challenge.
Saner et al sought to elucidate crucial points of these patients' management, since only a few hospitals are able to provide that service and not many critical care doctors therefore, are familiar with the nuances of managing a liver transplant patient postoperatively.
Therefore, I must congratulate Saner et al for the succinct points they make and the clarity of their review, which furthermore accomplishes to be easy to read, while simultaneously, being quite thorough....
The authors begin their review stating the most significant challenges in the postoperative period of liver transplantation, and thereafter, they analyze both the pathophysiology of the recipient as well as the suggested management. This review is highly organized throughout the text, and compels the reader to follow through, although, as stated already, this is not a common clinical scenario, at least not yet, when the number of the donors are not enough to satisfy the needs of the population...
Finally, another important point in this review, are the numerous references provided along with the main text, which encompass all the major literature on this topic and may guide the interested reader in pursuing more information about this complex situation....
Author Response
Dear Reviewer,
many thanks for these comments. As requested, a final spell and language check by a native a native speaker was done.
Reviewer 2 Report
This is a review on the postoperative care after liver transplantation.
I have some comments.
1. (L49) MELD needs explanation.
2. (L78) ECG needs explanation.
3. (L91) Is Takotsubo cardiomyopathy included? If not, it may be better to describe about Takotsubo cardiomyopathy.
4. (L176) The description on the viral infection should be added.
5. (L186) There is no subject in the sentence staring from “were used in 73% ..”
6. (L262) Is thrombotic microangiopathy included in thromboembolism? If not, it may be better to describe about thrombotic microangiopathy.
7. (L282) FFP needs explanation.
8. (L298~L366) The hepato-pulmonary syndrome and porto-pulmonary hypertension are the preoperative matter not post-operative. These sections should be omitted.
Author Response
Respected Reviewer 2
We appreciate your time and comments to improve the quality of the manuscript. All correction are highlighted in red color.
- (L49) MELD needs explanation.
- Is done in the text
- (L78) ECG needs explanation.
- Is done in the text.
- (L91) Is Takotsubo cardiomyopathy included? If not, it may be better to describe about Takotsubo cardiomyopathy.
- A paragraph describing Takotsubo is added
- (L176) The description on the viral infection should be added.
- Is also done
- (L186) There is no subject in the sentence staring from “were used in 73% ..”
- Is corrected
- (L262) Is thrombotic microangiopathy included in thromboembolism? If not, it may be better to describe about thrombotic microangiopathy.
- We added a paragraph about TMA
- (L282) FFP needs explanation.
- Is explained
- (L298~L366) The hepato-pulmonary syndrome and porto-pulmonary hypertension are the preoperative matter not post-operative. These sections should be omitted.
- The reviewer is right that HPS and PPH occurs preoperative. However, all these changes affects also the postoperative period. E.g., in 30% of cases HPS is not reversible, which makes the knowledge about that mandatory. The same for PPH. The reader should be informed about possible complications and their treatment. For that reason we would like to keep this paragraph.